# Light-Responsive Hexagonal Assemblies of Triangular Azo Dyes

**DOI:** 10.3390/molecules27144380

**Published:** 2022-07-08

**Authors:** Mina Han, Khin Moe

**Affiliations:** Department of Chemistry Education, Kongju National University, Gongju 32588, Korea; mireo192@gmail.com

**Keywords:** fluorescence characteristics, light response, molecular assembly, size-tunable hexagonal structures, triangular azo dye

## Abstract

The rational design of small building block molecules and understanding their molecular assemblies are of fundamental importance in creating new stimuli-responsive organic architectures with desired shapes and functions. Based on the experimental results of light-induced conformational changes of four types of triangular azo dyes with different terminal functional groups, as well as absorption and fluorescence characteristics associated with their molecular assemblies, we report that aggregation-active emission enhancement (AIEE)-active compound (**1**) substituted with sterically crowded *tert*-butyl (*t*-Bu) groups showed approximately 35% light-induced molecular switching and had a strong tendency to assemble into highly stable hexagonal structures with AIEE characteristics. Their sizes were regulated from nanometer-scale hexagonal rods to micrometer-scale sticks depending on the concentration. This is in contrast to other triangular compounds with bromo (Br) and triphenylamine (TPA) substituents, which exhibited no photoisomerization and tended to form flexible fibrous structures. Moreover, non-contact exposure of the fluorescent hexagonal nanorods to ultraviolet (UV) light led to a dramatic hexagonal-to-amorphous structure transition. The resulting remarkable variations, such as in the contrast of microscopic images and fluorescence characteristics, were confirmed by various microscopic and spectroscopic measurements.

## 1. Introduction

The facile construction of light-responsive fluorescent architectures with desired morphologies and functions has been an attractive research topic in the fields of chemistry, biology, biomedicine, and material science due to the potential applications of such architectures in sensing, bioimaging, drug delivery, memory, and molecular machines [1,2,3,4,5,6,7,8,9,10,11,12,13,14,15,16,17]. The rational design of small building block molecules and knowledge of their spatial arrangements, which are driven by non-covalent intermolecular interactions such as hydrogen bonding, van der Waals forces, π–π stacking, and hydrophobic interactions, are important in creating new stimuli-responsive organic nano- and microstructured materials [1,2,3,4,5,6,7,8,9,10,11,12,13,14,15,16,17,18,19,20,21,22,23,24,25,26,27,28,29,30,31,32,33,34,35,36,37]. The synthesis of photochromic organic compounds capable of regular molecular stacking while securing sufficient free space required for reversible light-induced conformational changes can be more effective than the dense packing of planar chromophores [38,39,40,41,42]. For example, in the case of sterically crowded azobenzene-containing molecules designed by considering the large distortion of benzene rings and the restriction of free rotation around single bonds, the reversible molecular switching can be monitored not only in solutions but also in the aggregated states, resulting in unique spectroscopic features and morphological changes [38,39,40,41,42,43,44,45,46].

Azobenzene is one of the most popular and widely used photochromic compounds, but it is known as a non-fluorescent π-conjugated dye at ambient temperature [47,48,49,50,51,52]. This is because the energy associated with the excited state is consumed by fast conformational changes instead of emission [53,54,55]. On the other hand, Gober and coworkers performed absorption and fluorescence spectroscopic studies of various azo dyes containing *ortho*-hydroxy groups in a wide range of solvents [56]. The −N = N− and O−H···N bonds in the molecule may contribute to light-induced *trans* ↔ *cis* conformational changes and intramolecular proton-transfer reactions, which are possibly responsible for fluorescence in solutions [56,57,58,59,60]. Lee et al. demonstrated that threefold-symmetric conjugated azo chromophores underwent conformational switching through torsional motions on the C–N bonds around the molecular core [61]. The thin-film samples exhibited reversible color switching upon exposure to selected chemical vapors. Han et al. reported that the introduction of flexible terminal groups into triangular azo dyes enabled high light-induced conformational changes (~90%) [11]. In addition, the triangular compounds tended to assemble into fluorescent microspheres and formed soft mesophase structures in the bulk. However, the construction of light-responsive nanostructures with regular molecular stacking, such as rods, sticks, and cubes, has not yet been accomplished.

Based on preliminary investigations of light-induced conformational changes of triangular dyes with different terminal functional groups, their molecular assemblies, and aggregation-induced emission enhancement (AIEE [62,63,64,65,66,67]) properties (Figure 1), in this paper, we describe that compound **1** substituted with sterically crowded *tert*-butyl (*t*-Bu) groups grew into highly stable hexagonal rods with AIEE characteristics. It was possible to tune the sizes of the fluorescent hexagonal structures from nanometer-scale short rods to micrometer-scale long sticks by increasing the initial concentration (Figure 2). Moreover, non-contact light irradiation induced prominent morphological transformation from hexagonal to amorphous structures, which was accompanied by noticeable variations, such as the contrast in the microscopic images and fluorescence characteristics.

## 2. Results and Discussion

Information about the effect of terminal substituents linked to triangular azo chromophores (compounds **1**–**4**, Figure 1) is significant for understanding the important factors affecting both light-induced molecular structure changes and the construction of self-assembled architectures with desired shapes and functions. Two triangular molecules with electron-donating triphenylamine (TPA, compound **3** [68]) and electron-withdrawing bromo (Br, compound **4** [69]) groups have low solubility in solvents such as toluene, chloroform, dichloromethane, and tetrahydrofuran (THF). By comparison, compounds **1** and **2** are soluble in common organic solvents. Whereas compounds **3** and **4** rarely exhibited photoisomerization, the exposure of the compound **1** sample to ultraviolet (UV) light resulted in approximately 35% molecular switching from *C*_3_-symmetric to asymmetric structures, as evidenced by ^1^H NMR experiments (Appendix A). In addition, compound **2** substituted with flexible *n*-butyl groups at three terminals underwent photoisomerization as high as approximately 90% [11]. These experimental results suggest that the flexibility of terminal functional groups, rather than their electronic nature and steric effect, is an essential factor in determining light-induced conformational changes.

Interestingly, the introduction of such terminal substituents enables the control of the morphologies of molecular assembled structures and their AIEE characteristics. For instance, compound **4** forms one-dimensional (1D) fibrous structures, regardless of the solvent polarity [69]. By contrast, compound **2** has a strong tendency to assemble into spherical objects in THF-H_2_O mixed solutions [70], whereas the substitution of sterically crowded *t*-Bu groups leads to the formation of hexagonal structures such as short nanorods and long sticks. Notably, after the molecular assemblies, the fluorescence efficiencies for **1** and **2** improved by approximately 2–3%. However, compound **3** did not exhibit an AIEE property, which is probably associated with intramolecular proton-transfer reactions and resonance energy transfer [71,72,73]. From these results, it can be inferred that rather than the electronic nature of the substituent, both the significant distortion around the central rings and the suppression of intramolecular rotation around the single bonds play key roles in determining the AIEE.

### 2.1. Morphology and Size Control of Hexagonal Structures

To gain insight into the morphology and size control of assembled structures, we prepared compound **1** THF solutions with concentrations ranging from 10 μM to 1000 μM. As ultrapure water was slowly added to each solution, the resulting mixtures became milky and opaque regardless of the concentration. The mixtures were allowed to stand for two days for sufficient molecular assembly. Whereas the turbidity of dilute mixtures remained stable for several days to one month or more, the concentrated mixtures gradually produced precipitates, which sank to the bottom of the quartz cell. The concentration dependence of the turbidity and precipitation suggests variations in the morphologies and dimensions of the assembled structures. Our scanning electron microscopy (SEM) measurements indicate that nanometer-scale puffcorn-shaped structures with rounded edges were mainly found in 2 μM THF-H_2_O mixtures (Figure 2b and Figure 3a–c). Their width and length were ~100 and ≤300 nm, respectively. The reason that molecular assembly stopped at roundish puffcorn-shaped structures rather than at angled hexagonal rods was that, upon adding excess water to the dilute THF solution (1/4, *v*/*v*), a large number of nuclei formed in a very early stage. However, due to their limited number, the triangular azo dyes could not develop sufficiently via regular molecular stacking. Whereas the melting point of hexagonal structures was 198–205 °C, the puffcorn-shaped structures completely melted in the 143–150 °C range. This lower melting temperature supports that the round puffcorn-shaped structures were formed via relatively loose molecular stacking.

Solid hexagonal rods appeared in the samples with concentrations ranging from 10 μM to 50 μM (Figure 3d–i and Appendix A). The melting temperatures (195–200 °C) of the rods obtained from the 10 μM mixtures were about 50 degrees higher than that of the puffcorn-shaped structure. As the concentration was increased to 50 μM, the average width and length increased to ~200 and ~520 nm, respectively. The rods melted at 198–204 °C. These experimental results suggest that the samples with concentrations of 10 μM or more contained significant numbers of triangular molecules that could be regularly stacked to form hexagonal structures with increased crystallinity.

In THF-H_2_O mixtures with concentrations of 83 μM or higher, compound **1** tended to grow into relatively long sticks with an average width and length of ~300 nm and ~2.7 μm, respectively, rather than growing into short nanorods (Figure 2 and Figure 3j–o). The microsticks were fragile and broke into two or more pieces during the sample preparation process. The magnified SEM image provided in Figure 3k shows that the broken cross-section retained a hexagonal shape. Moreover, even in the most concentrated THF-H_2_O mixture (500 μM, Figure 3m–o), microstick growth stopped at approximately seven micrometers. That is, compound **1** did not develop into elongated fibrous structures.

On the other hand, switching from THF to more nonpolar solvents, such as toluene and dichloromethane, promoted the formation of elongated fibers, and their length reached several hundred micrometers, irrespective of the concentrations (Appendix A). That is, in non-polar solvent environments, the periphery of the polar central rings in triangular molecules appeared to be more planar through intramolecular hydrogen bonds [74]. Accordingly, the relatively planar molecules preferentially appeared to pile up on the surface of the already formed crystal nuclei, eventually developing into elongated fibrous structures.

### 2.2. Growth into Fluorescent Hexagonal Nanorods

Next, we investigated the growth process into fluorescent hexagonal nanorods at a fixed concentration. Based on the above-mentioned experimental results, we used a 10 μM THF-H_2_O mixture containing a sufficient number of molecules to form angled nanorods. The SEM images in Figure 4 display the morphological evolution into hexagonal rods as a function of the storage time of the opaque mixture. In a very early stage, flat egg-shaped and peanut-shaped aggregates of approximately 50–100 nm in size were frequently found in the sample (Figure 4a). When the red-pink turbid mixture was stored for two hours, the suspension contained bowling pin-shaped structures (Figure 4b) and flat objects with sizes of 50–150 nm. Intriguingly, the inset SEM images indicate that one end of the bowling pin-shaped structure was a narrow triangle with a width of 50–100 nm; the other was a relatively wider hexagon with a width of 50–150 nm. This unique structure appears to be an intermediate structure produced in the process of triangular molecules growing into angled hexagonal nanorods. A storage time of two days was sufficient for growth into angled hexagonal rods (Figure 4c). Surprisingly, even after storage for a long period of two months, the mixture produced few observable precipitates and the size of hexagonal rods hardly changed (Figure 4d).

Figure 5a shows the absorption spectra measured as a function of storage time. The two strong absorption bands centered at 378 and 502 nm [38,40,56,60] were red-shifted to 400 and 517 nm, respectively, after hexagonal assembly. At the same time, the as-prepared orange transparent solution turned into a red-pink turbid mixture (inset photo in Figure 5a). Even after storage for two days, the spectral shape and the absorbance at 700 nm (indicated as turbidity) did not change significantly. Upon excitation with green light, bright red fluorescence was detected, with a maximum fluorescence wavelength near 645 nm, which was red-shifted by 45 nm relative to the orange transparent solution (Figure 5b). Additionally, the red fluorescence intensity before and after hexagonal assembly improved roughly 20-fold, indicating that compound **1** was AIEE-active.

The X-ray diffraction (XRD) pattern of the fluorescent hexagonal structures is shown in Figure 6. Compared to the XRD profile of soft microspheres assembled from compound **2** [11], the microsticks exhibited better crystallinity. The strong peak appearing at 2*θ* = 4.28° (*d* = 2.06 nm) matches the length of one side of the triangular molecule (Figure 1). The peak at 2*θ* = 10.01° (*d* = 0.88 nm) presumably corresponds to the distance between two phenyl rings via an azo group. In addition, the small but characteristic peaks at 2*θ* = 21.78° and 23.06° (*d* = 0.40 and 0.38 nm) in the wide-angle region are somewhat longer than the frequently reported interlayer distances between planar aromatic cores [75,76,77,78]. Based on these experimental results, we can infer the following. The large distortion around the molecular center and the sterically crowded terminal substituents probably caused (i) significant restriction to free rotation around single bonds and (ii) relatively weak intermolecular interactions. Hence, hexagonal assembly via regular molecular stacking seems to proceed while maintaining a large volume per molecule. For this reason, it is expected that light-induced *C*_3_ symmetry breaking is possible in the aggregated state and, as a result, the degree of crystallinity of the hexagonal structure decreases, thereby causing not only morphological deformation [79,80,81,82,83] but also variations in fluorescence characteristics.

### 2.3. Light-Induced Hexagonal-to-Amorphous Transition

To corroborate the light-induced morphological transformation of the hexagonal structures, nanorods were placed on quartz (or glass) substrate. Upon excitation with green light, bright red fluorescence was observed with a maximum fluorescence wavelength near 650 nm, which was red-shifted by ~50 nm relative to the transparent THF solution of the same concentration (Figure 7f). When the sample was exposed to UV light, the shape of the hexagonal rods was noticeably deformed. Further light irradiation eventually melted the rods (Figure 7c). At the same time, the red fluorescence intensity decreased dramatically (Figure 7d−f). More interestingly, upon excitation with UV light (Figure 7g−i), instead of the significant reduction in red fluorescence intensity, the light-exposed regions had very weak blue fluorescence near 460 nm (fluorescence quantum yield ~0.1%). This interesting phenomenon can be interpreted as follows. If the number of conformational and tautomeric species produced by light-induced *C*_3_ symmetric breaking increases, intermolecular interactions would considerably weaken, resulting in a crystalline-to-amorphous structure transition. Consequently, the chemical species forming the amorphous structures are considered to be responsible for the very weak blue fluorescence.

### 2.4. Hexagonal-to-Amorphous Transition Accompanied by Prominent Changes in Fluorescence Characteristics and Good Contrast of Microscopic Images

The hexagonal structures can be used in a variety of applications, such as non-contact optical data storage and sensing systems. For instance, when fluorescent nanorods placed on quartz substrate were exposed to light (100 mW cm^−2^, 30 min) through a photomask in which numbers and horizontal lines were recorded, it was possible to store information with excellent contrast and good resolution. First, the optical microscopy (OM), and fluorescence optical microscopy (FOM) images demonstrate that while the hexagonal rod structures were dark red, their color became significantly lighter when exposed to light (Figure 7b and Figure 8). Instead of a striking reduction in red fluorescence intensity, the area exposed to light exhibited very weak blue fluorescence (Figure 8b). Thus, the recorded information was observable with a different fluorescence color when excited with not only green light but also UV light.

In addition to the OM and FOM results, the SEM images in Figure 8d,e were measured by magnifying the brighter and darker areas of Figure 8c, respectively. The hexagonal rods that were not exposed to light retained their shapes and looked brighter. In contrast, the light-exposed rods partially melted and looked darker. Hence, the information was recorded with good resolution in a non-contact manner. The light responses of the hexagonal rods were successfully demonstrated by various measurement methods, including fluorescence spectroscopy, OM, FOM, and SEM.

## 3. Conclusions

A triangular azo dye with sterically crowded functional groups grew into size-tunable hexagonal structures ranging from flat structures and nanorods to microsticks, depending on the concentration. Notably, the fluorescent nanorods obtained from dilute mixed solutions remained very stable for periods from several days to more than a month. Our study on four types of triangular molecules with different terminal functional groups demonstrates that large distortion around molecular centers, as well as the balance between the steric effects and the flexibility of the terminal substituents, is important in determining the following properties and functions: (i) light-induced conformational changes of monomeric species, (ii) morphologies of self-assembled structures, (iii) AIEE characteristics, (iv) light-induced morphological transformation, and (v) resulting remarkable changes in fluorescence characteristics and good contrast in microscopic images. The findings of this work provide possibilities for designing new artificial self-assembling systems as well as achieving novel photo-functional organic nanomaterials.

## 4. Materials and Methods

### 4.1. Synthesis of Compound **1**

(2,4,6-tris{(E)-(4-t-butylphenyl-2,6-diethylphenyl)diazenyl}benzene-1,3,5-triol (**1**, Figure 1) was prepared by Suzuki coupling reaction of precursor 2,4,6-tris((E)-(4-bromo-2, 6-diethylphenyl) diazenyl) benzene-1,3,5-triol in the presence of a palladium (0) catalyst. A catalytic amount of tetrakis(triphenylphosphine)palladium (0) was added to a solution of precursor 1 (0.30 g, 0.35 mmol) in toluene (40 mL). 4-*t*-Butyl phenyl boronic acid (0.25 g, 1.4 mmol) and a solution of NaHCO_3_ in water (1N, 30 mL) were added to the above mixture. The mixture was stirred at 110 °C for five days. After the mixture was cooled to room temperature, water and chloroform were added. The organic layer was separated and purified by silica gel column chromatography (n-hexane: chloroform, *v*/*v* = 1/1) to yield a dark red solid (0.18 g, yield: 51%).

^1^H NMR (400 MHz, CDCl_3_, δ): 7.60 (*d*, 6H, *J* = 8.4 Hz, aromatic), 7.52 (*d*, 6H, *J* = 8.4 Hz, aromatic), 7.42 (*s*, 6H, aromatic), 2.96 (*q*, 12H, *J* = 7.6 Hz, ArC*H*_2_CH_3_), 1.39 (s, 27H, ArC(C*H*_3_)_3_), 1.36 (*t*, 18H, *J* = 7.6 Hz, ArCH_2_C*H*_3_). ^13^C NMR (900 MHz, CDCl_3_, d): 178.63, 150.71, 140.61, 138.02, 137.61, 136.63, 129.39, 126.75, 126.44, 125.78, 34.559, 31.37, 25.71, 14.97. FAB-HRMS (m/z): [M + H] calculated for C_66_H_79_N_6_O_3_ = 1003.6214; found: 1003.6217 (=M + 1).

### 4.2. Materials

Spectroscopic grade organic solvents (tetrahydrofuran (THF), toluene, and methanol (MeOH)) were purchased from Kanto Chemical Co., INC, Tokyo, Japan. Ultrapure water (which was purified to reach a minimum resistivity of 18.0 MΩ·cm (25 °C) using a μPure HIQ water purification system, Romax, Hanam, Korea) was utilized in all experiments.

### 4.3. Characterization

After a 30 s nitrogen purge, a screw-cap quartz cuvette containing azo dye solution was sealed with Parafilm^®^. UV-vis absorption and fluorescence spectra were obtained using a Shimadzu UV-2600 UV-vis spectrophotometer (Shimadzu Corporation, Kyoto, Japan) and a Horiba FluoroMax-4 spectrofluorometer (Horiba Ltd., Kyoto, Japan), respectively. ^1^H and ^13^C NMR measurements were carried out with a Bruker 400 MHz spectrometer. The azo dye solutions and nanorods were exposed to UV light (Tokina Supercure-204S, Tokyo, Japan), generated by a combination of Toshiba color filters: UV-35 + UV-D36A. The solution samples were exposed to low light intensities (1.5–2.0 mW cm^−2^). Optical microscopy (OM) and fluorescence optical microscopy (FOM, λ_ex_ = 520–550 nm) experiments were performed using an Olympus BX53 microscope after placing two to three drops of a mixed suspension onto clean glass or quartz substrate. Field-emission scanning electron microscopy (FE-SEM; HITACHI SU8020 (Hitachi High-Technologies Corporation, Tokyo, Japan) and TESKAN-MIRA3-LM (TESCAN ORSAY HOLDING, a.s., Brno, Czech Republic) samples were coated with approximately 5–10 nm thick platinum layers using a Cressington 108 Auto Sputter Coater (Ted Pella, Inc., Redding, CA, USA). X-ray diffraction (XRD) data were collected with Cu Kα radiation on Rigaku R-AXIS-IV and R-AXIS-VII X-ray imaging plate detectors to determine the structure of the triangular molecule.

## Data Availability

The data are included within the manuscript and Appendix A.

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
