# Peer review of "Light-Responsive Hexagonal Assemblies of Triangular Azo Dyes"

_molecules, 2022, doi:10.3390/molecules27144380_

Round 1

Reviewer 1 Report

In this manuscript, the authors reported novel crystalline, hexagonal assemblies of triangular Azo dyes, and tested their light responsive properties including AIEE and crystal to amorphous structure transition. The self-assembly of the dyes were systematically characterized by different imaging and spectroscopy techniques, with clear data analysis and interpretation. I would recommend its publication after addressing the following comments /questions:

1.     In Fig. 6, the XRD peaks can match the length of the triangular molecule and some interlayer distances. Since here it is a more crystalline structure of clear hexagonal morphology, could the authors be possible to make some hypothesis of the molecular structure of the self-assembled rods (which match the XRD patterns)? Fig. 2a showed that they will have stacking into a hexagonal shaped dimer, and it would be instructive to the readers to have some idea of how they will further assemble. With this the following descriptions on the steric and rotation will be also easier to follow.

2.     It is interesting phenomena that in Fig. 7b, the hexagonal rods merged together upon exposure to UV light. Basing on experimental setup described in the materials and methods session, is it possible to be caused by the drying of the droplet? If the UV light is strong and shine on the sample for long time, the reviewer think it might be possible that the exposed region is heated up faster, has solvent evaporation and thus some local “coffee ring” effect. In this case, sandwiching the droplet sample between two quartz plates could be a better choice for giving more reliable UV exposure study, without affecting temperature and local concentration. Also, it would be helpful for the authors to give more detailed information on this experimental such as light intensity and time scale.

3.     On line 220, the authors mentioned that the light-exposed rods looked melted and darker, could the observed melting actually happen during SEM imaging instead of during UV exposure, if the rods can’t be coated well by platinum after exposure?

Minor corrections:

1.     In Fig. 2a, the “solvent polarity” label is not showing clear trend

2.     In Fig. 2b, it would be helpful to label what the black and orange circles represent

3.     In Fig. 3d, the scale bar is very hard to read

4.     The units in Fig. 6 should be nm instead of Å. All the degree symbols in the manuscript are recommended to change to the circle on the right top corner (°) rather than in the middle.

Author Response

RE: Manuscript ID: molecules-1799891 entitled " Light-Responsive Hexagonal Assemblies of Triangular Azo Dyes" by Mina Han and Khin Moe

Dear Reviewer 1:

We are sending herewith our revised manuscript for publication in Molecules.

Two important changes in the manuscript include provision of the following information in the Experimental section and addition of Supplementary Materials Figure S1B.

We have addressed Reviewer’s important comments and summarized our response in the attached page. We appreciate the Reviewer's valuable comments.

I am looking forward to your response.

Sincerely

Mina Han, Associate Professor

E-mail: hanmin@kongju.ac.kr

Attached: (1) Revised manuscript (doc & pdf), (2) Supplementary Materials (pdf), (3) GA, (4) Response to Reviewer’s comment.

Reviewer 2 Report

In this manuscript the authors report on self-assembled morphologies and optical properties of a series of triangular azo compounds. The four compounds 1-4 exhibit different solubility to organic solvents and afford unique nano-to-micro structures of molecular assemblies, such as fibers, puffcorns, rods, sticks, hexagonal, and so on. Especially, the compound 1 shows interesting growth behavior of hexagonal rods, depending on the concentration and the storage time. The self-assembled structures are well characterized by SEM and XRD. Also, the compound 1 exhibits the significant AIEE property and the molecular aggregates emit bright red fluorescence. Moreover, the material undergoes photoinduced hexagonal-to-amorphous transition along with the switching of morphology and fluorescence. The observation of the unique self-assembled morphologies and photoresponses of the organic dyes in the present work is of importance in the design and development of photofunctional materials and devices and would attract attention of researchers in the wide fields. In my opinion, the paper is acceptable for publications in Materials after minor revision.

1) Page 3, line 80 “..., exposure of compound 1 sample to ultraviolet (UV) light results in approximately 35% molecular switching from C3-symmetric to asymmetric structures, as evidenced by 1H NMR experiments (Figure S1).”

Does the compound 1 show any absorption spectral changes in dilute (non-aggregated) solutions upon photoisomerization? The experimental data of absorption spectral changes should be shown in the main text or the supplementary material.

2) Page 7, line 194 “More interestingly, instead of the significant reduction in red fluorescence intensity, the light-exposed regions had very weak blue fluorescence near 460 nm (fluorescence quantum yield ~0.1%), upon excitation with UV light (Figure 7g→i).  ...  Consequently, the chemical species forming the amorphous structures are considered to be responsible for the very weak blue fluorescence.”

The blue fluorescence near 460 nm under excitation with UV light corresponds to the optical transition with higher energy (at shorter wavelength) than S1-S0 fluorescence. Is it ascribed to S2-S0 (so-called anti-Kasha) fluorescence? The origin of the blue fluorescence should be discussed in more detail.

Author Response

RE: Manuscript ID: molecules-1799891 entitled " Light-Responsive Hexagonal Assemblies of Triangular Azo Dyes" by Mina Han and Khin Moe

Dear Reviewer 2:

We are sending herewith our revised manuscript for publication in Molecules.

Two important changes in the manuscript include provision of the following information in the Experimental section and addition of Supplementary Materials Figure S1B.

We have addressed Reviewer’s important comments and summarized our response in the attached page. We appreciate the Reviewer's valuable comments.

I am looking forward to your response.

Sincerely

Mina Han, Associate Professor

E-mail: hanmin@kongju.ac.kr

Attached: (1) Revised manuscript (doc & pdf), (2) Supplementary Materials (pdf), (3) GA, (4) Response to Reviewer’s comment.
